# Experimental Study of White Light Interferometry in Mach–Zehnder Interferometers Based on Standard Single Mode Fiber

**DOI:** 10.3390/s24103026

**Published:** 2024-05-10

**Authors:** José Luis Cano-Perez, Jaime Gutiérrez-Gutiérrez, Christian Perezcampos-Mayoral, Eduardo L. Pérez-Campos, María del Socorro Pina-Canseco, Lorenzo Tepech-Carrillo, Marciano Vargas-Treviño, Erick Israel Guerra-Hernández, Abraham Martínez-Helmes, Julián Moisés Estudillo-Ayala, Juan Manuel Sierra-Hernández, Roberto Rojas-Laguna

**Affiliations:** 1Facultad de Medicina y Cirugia, Universidad Autónoma “Benito Juárez” de Oaxaca (FMyC-UABJO), Ex Hacienda de Aguilera S/N, Calz. San Felipe del Agua, Oaxaca de Juárez C.P. 68120, Mexico; cpecmay@uabjo.mx (C.P.-M.); mpina.cat@uabjo.mx (M.d.S.P.-C.); 2Facultad de Sistemas Biologicos e Innovacion Tecnologica, Universidad Autónoma “Benito Juárez” de Oaxaca (FASBIT-UABJO), Av. Universidad S/N, Ex-Hacienda 5 Señores, Oaxaca de Juárez C.P. 68120, Mexico; ltepech@uabjo.mx (L.T.-C.); mvargas.cat@uabjo.mx (M.V.-T.); ghernandez.cat@uabjo.mx (E.I.G.-H.); 3National Technology of Mexico/IT Oaxaca, Oaxaca de Juárez C.P. 68033, Mexico; pcampos@itoaxaca.edu.mx; 4Facultad de Contaduría y Administración, Universidad Autónoma “Benito Juárez” de Oaxaca (FCA-UABJO), Av. Universidad S/N, Ex-Hacienda 5 Señores, Oaxaca de Juárez C.P. 68120, Mexico; amartinez@uabjo.mx; 5Division de Ingenierias, Campus Irapuato-Salamanca, Universidad de Guanajuato, Carretera Salamanca-Valle de Santiago km 3.5 + 1.8, Comunidad de Palo Blanco, Salamanca C.P. 36885, Mexico; julian@ugto.mx (J.M.E.-A.); jm.sierrahernandez@ugto.mx (J.M.S.-H.); rlaguna@ugto.mx (R.R.-L.)

**Keywords:** white light interference, bi-tapers and core-offset Mach–Zehnder interferometer, temperature sensor

## Abstract

In this work, we experimentally analyzed and demonstrated the performance of an in-line Mach–Zehnder interferometer in the visible region, with an LED light source. The different waist diameter taper and asymmetric core-offset interferometers proposed used a single-mode fiber (SMF). The visibility achieved was V = 0.14 with an FSR of 23 nm for the taper MZI structure and visibilities of V = 0.3, V = 0.27, and V = 0.34 with FSRs of 23 nm, 17 nm, and 8 nm and separation lengths L of 2.5 cm, 4.0 cm, and 5.0 cm between the core-offset structure, respectively. The experimental investigation of the response to the temperature sensor yielded values from 50 °C to 300 °C; the sensitivity obtained was 3.53 a.u./°C, with $R^{2}$ of 0.99769 and 1% every 1 °C in the transmission. For a range of 50 °C to 150 °C, 20.3 pm/°C with a $R^{2}$ of 0.96604 was obtained.

## 1. Introduction

The technique of optical interferometry is widely used in the fabrication of fiber optic sensors due to its accurate measurement of the physical quantities of industrial and medical fields. Optical interferometry is based on the interference of two or more beams from a light source launched into dielectric media with different optical length paths, creating an optical phase difference between the beams and obtaining an interference pattern [1]. Small changes in the optical path induce variation in the interference light intensity to the output sensor, and one can obtain information about these changes for the measurement of physical parameters. The optical fiber sensor based on an in-line Mach–Zehnder interferometer (MZI) has become an attractive research subject due to its simplicity, adaptability, manufacturing, compact size, and stability [2]. They are usually fabricated using single-mode fibers (SMF)s [2,3,4,5,6,7], multimode fiber (MMFs) [8,9,10,11,12,13,14], photonic crystal fibers [15,16,17], and erbium-doped fibers (EDFs) [8,18] with different configurations: core-offset [2,3,4], tapers [5,6,7,18], two-peanut-shape [8,19], and microcavities [19,20,21], among others. The MZIs are generally used to measure several physical variables, such as curvature [3,18], refractive index [3,5,6,7], temperature [5,19], strain [15], and gas sensing [17]. The optical fiber MZIs have been mainly studied in the infrared (IR) [1,2,3,4,5,6,7,8,9,14,15,16,17,18], near-infrared (NIR) [20,21], and visible spectrum (VIS), in contrast to the design of interferometric sensors based on optical fibers in the VIS region [19,20,21]. The NIR and VIS interference using optical fibers SM800 and SM600 have been researched by Eftimov et al. [20,21], and they used optical fibers SM800 and SM600. The spectrum interference was obtained in the range of 670 nm to 1100 nm. The interference emission in the VIS region was obtained by a taper MZI based on microfibers or nanofibers (MNFs). The MZI was fabricated with two MNFs placed on a ${\mathrm{MgF}}_{2}$ substrate [22]. The supercontinuum source was used in their experimental setup [19,20,21].

In this work, during experimentation, the proposed in-line MZIs demonstrated an emission pattern of interference with white light from an LED source. For visible light interference research, two MZIs with an SMF-SMF-SMF structure were fabricated. The first interferometer was bi-taper and the second was based on the core-offset splicing technique. Both MZIs were fabricated using standard optical fibers (SMF-28). The interferences emissions were analyzed individually.

## 2. Design and Fabrication of Mach–Zehnder Interferometer

Two interferometers were proposed and fabricated for white light analysis interference, taper, and core-offset in-line MZI. 

### 2.1. Design and Fabrication of Taper MZI

The tapered MZI was fabricated using a stretching machine designed by the Universitat de València. It simultaneously heats and stretches a short section of single-mode fiber (SMF-28 of Thorlabs, Newton, NJ, USA, with 8.2 µm and 125 µm core and cladding diameters, respectively, and an operating wavelength range of 1260 nm to 1650 nm). The flames were generated with a precise mixture of butane and oxygen gas. The section of SMF was fixed at both ends by two holders on a servomotor. In the stretching machine, the diameter of the MZI waist was controlled by original software on a PC; these data were the only set required for its manufacturing (Figure 1).

The region of fiber created by the stretching machine had a small and uniform waist, where the fiber diameter changed as a result of the stretching, fabricating an adiabatic tapered MZI [6].

Figure 2 shows the schematic of a bi-taper MZI which consists of a single-mode fiber and can be divided into first taper T_1_ and second taper T_2_, with a separation distance L. The tapers T_1_ and T_2_, can be divided into regions: down taper (D_down1_ and D_down2_), waist length (L_1_ and L_2_), waist diameter (W_1_ and W_2_), and up-taper (D_up1_ and D_up2_), respectively: D_1_ = D_down1_ = D_up1_ and D_2_ = D_down2_ = D_up2_. The tapers T_1_ and T_2_ can be divided into regions: down and up-taper (D_1,2_), waist length (L_1,2_), and waist diameter (W_1,2_), respectively. In an SMF, coherent light propagates along the core and the fundamental mode (*LP*_01_) is generated, but when a low-coherence source is launched into the SMF, higher-order modes (*LP*_0m_) are excited [11,12]. The interference effects may exist between different core modes or between different cladding modes of the MZI structure. The modal interference between modes of the MZI structure can be given by [11,12,13]:(1)$$I=I_{1}+I_{2}+2\sqrt{I_{1}*I_{2}}\cos\left(\theta \right)$$
where $I_{1}$ and $I_{2}$ are the intensities of the interference signal, the core modes, and the cladding modes, respectively. θ is the phase difference between the core mode and the cladding mode, and is given by:(2)$$\theta =\frac{2\pi \;\left(\Delta n_{eff}\right)L}{\lambda }$$
where λ is the wavelength of the light source, L is the fiber waist length of the MZI, and the $\Delta n_{eff}=n_{eff}^{core}-n_{eff}^{clad}$ is the refractive index difference of the two interferometer arms; $n_{eff}^{core}$ and $n_{eff}^{clad}$ are the effective refractive index of the core and cladding mode of the fiber, respectively. When the interference signal reaches its minimum at θ=π(2m+1) in Equation (1), the wavelength of the mth order attenuation peak is written as in Equation (2). As a result, the free spectral range (FSR) of such a fiber interferometer is expressed as [8,16]:(3)$$FSR=\frac{\lambda^{2}}{\Delta n_{eff}}L$$

Several MZIs with different lengths L and waist diameters were fabricated, and their transmission spectra were analyzed.

The experimental setup for analyzing the pattern interference in the VIS region of MZIs is shown in Figure 3 as an LED source (DLF, Model LP3WBCD, Guangdong, China, power of 3 Watts, natural white) with a wavelength range of 400 nm to 650 nm (the spectral emission at room temperature is shown in Figure 4). The light was introduced into fiber optics with an Olympus Plan N 20×/0.40 micro-scope objective, Nanjing, China, which was mounted on fiber alignment 3-axis flexure stages (MBT610/N of Thorlabs, Newton, NJ, USA); then, the light traveled through in MZI and the spectral emission interference was obtained using a Spectrum Analyzer (Ocean Optics, USB650, Boston, MA, USA) with a slit of 5 μm in the software configuration of the computer.

Table 1 shows the results from a collection of samples of MZIs with different waist diameter parameters for $T_{1}$ and $T_{2}$ from 10 µm to 60 µm. The separation distance between two bi-tapers is kept to L=3 cm and the waist length between 1.83 mm to 29.35 mm. The software requires the waist diameter, and the other parameter is automatically calculated.

### 2.2. Design and Fabrication of Core-Offset MZI

Figure 5a shows a U-type MZI. The MZI was fabricated using SMF-28 (Thorlabs) with core and cladding of 8.2 µm and 125 µm, respectively. The schematic diagram of the experimental setup is similar to Figure 3. For the manufacturing process of the U-type MZI, three segments of SMF and a section of fiber length L were used between two SMFs, joined by a Fitel S178A fusion splicer, Tokyo, Japan. First, a 70 cm section was cut with the fiber cleaver and fixed with masking tape to a flat aluminum bar, leaving approximately 7 cm of optical fiber at one end of the bar; see Figure 5b. It was carefully placed in the left holder of the fusion splicer. The flat aluminum bar was placed on top of the lab jack of vertical travel L490 at the same height as the holders on the fusion splicer. Second, another optical fiber segment was placed in the right holder, aligning the two core fibers. Afterward, the right holder was displaced in the $\left[-x_{1},0.0\right]$ axis direction, the splice was made in manual mode, and the SMF-1300 program setting was used to splice with one discharge. Thirdly, the spliced fiber was removed and the second section of the interferometer length L was cleaved. A ceramic fiber scribe CSW12-5 was used when the length L was less than 4 cm, taking care not to break it. The cleaved part of the interferometer with length L was placed in the left holder and fusion-spliced with another segment of 70 cm of fiber, and then aligned in the $\left[+x_{2},0.0\right]$ axis direction. The same configuration was used for the fusion splicer. In order to analyze the influence of the offset distance on the propagation of the core and cladding modes, it was necessary to fabricate and test several symmetrical and asymmetrical interferometers, with length of 2.5 cm, 4 cm, and 5 cm; see Table 2.

Table 2 shows a collection of samples of core-offset MZIs with different length segments of L = 2.5 cm, 4 cm, and 5 cm. For the analysis of spectrum emissions, the MZIs were fabricated using different displacement on the x, y axis.

## 3. Results

The interference pattern was measured in tapered and core-offset MZIs. The results are presented in two parts.

### 3.1. Analysis of Tapered MZI

Figure 6a shows the spectral emissions, and Figure 6b shows the spatial frequency that are obtained by Fourier transform. In Figure 6a, the interference emission obtained for the MZI with a symmetrical waist length relation of ([45-45]) μm is weak. For this case, we can see in Figure 6b that the interference pattern output was formed by the fundamental core mode and three cladding modes at p~44 nm, p~24 nm and p~17 nm; these spatial frequencies were generated by the multimode fiber of the MZI [10,11,12], and there were more higher-order modes that contributed weakly to forming the interference pattern with an incompletely sinusoidal waveform. The best sinusoidal interference emission profile was obtained with an asymmetric waist length relation of 10–20 μm. Figure 6b shows that the interference was formed by the contribution of the fundamental core mode and a strong cladding mode in p~24 nm. The period signal was rationed between spatial frequency and period sinusoidal components using ν=1/p [18].

The spatial frequency analysis was similar to an adiabatic [6,17], because the interference pattern of the fabricated asymmetric tapers was mainly formed by the fundamental core mode $I_{1}$ and cladding mode $I_{2}$. The asymmetric and symmetric MZIs with waist length relations of [50-60], [15-10], and [10-10] μm (see Figure 6a) did not have an interference pattern due to their weak intensity of the cladding modes.

The spectral profile of the interferometer with a waist length relation of 10–20 µm is plotted in the Figure 7. The spectral response shape is a sinusoidal waveform, with the typical interference emission of a relation MZI [1]. The interference patterns are defined by the optical path difference, and its amplitude is a function of the fringe visibility or contrast (V) and is defined based on coherence theory [1,23]:(4)$$V=\frac{I_{Max}-I_{Min}}{I_{Max}+I_{Max}}$$

$I_{Max}$ is the maximal and $I_{Min}$ is the minimal intensities of wave oscillations. The fringes of interference with good visibility *V*, took values of 0.1≤V≤1. We obtained a visibility of *V* = 0.14, a FWHM = 13 nm, and FSR = 23 nm; moreover, for the other waist length relation MZI, the interference was weak or null. This parameter V is very important for optical fiber sensor application [1,23].

### 3.2. Analysis of Core-Offset MZI

The core-offset technique was studied experimentally with different displacement in the [x,y] axis, in both the first and the second fusion splice, to fabricate the MZI while keeping the length L at 4 cm. First, the MZIs numbered from #2 to #9 in Table 2 were analyzed. Figure 8a–c show the evolution of interference spectrum dependent on the displacement of the [x,y] axis in the splicing fiber. We can see that the extinction ratio (ER) increased or decreased as the displacement [*x, y*] of the MZIs increased. 

The spatial frequencies of the MZIs mentioned above are plotted in Figure 8d. For the MZI (#5), a strong cladding mode was present in p≈15 nm and contributed significantly to the emission pattern interference; therefore, the interference pattern was due to the fundamental core and cladding mode at p≈15 nm. For the MZIs (#6) and (#7), the amplitude of the spatial frequency in p≈15 nm was smaller by 75% proximally compared to the #5 MZI, but the output pattern also had a sinusoidal form, as can be seen in Figure 8b. 

We can see that the interference patterns for other core-offset MZIs are not completely sinusoidal waveform, as they were formed using the fundamental core mode and a number of high-order cladding modes. This was generated when the white light was launched into the an SMF; higher-order modes (LP_0m_) were excited [11,12]. It could be observed in all MZIs that the interference pattern was generated by the fundamental core mode and high-order cladding modes because the SMF fiber had a performance multimode. However, in MZI (#5), it can be observed that high-order cladding modes and higher-order modes were not predominant due to their weakness We found that the optimal displacement in the [x,y] axis for the MZI was [−4.6, 0.0] μm and [+4.6, 0.0] μm for the first and second junction splice, respectively.

Figure 9 shows the spectra emission outputs of the #1, #5, and #10 MZIs, with displacements in the [*x, y*] axis of [−4.6, 0.0] µm and [+4.6, 0.0] µm at the first and second fusion splice junction, respectively. We can see the output interference emission of the #1 MZI, Figure 9a, with a visibility of V = 0.3, the FWHM = 13 nm and an FSR = 23 nm, for the #5 MZI, we obtained a *V* = 0.27, FWHM = 8 nm, and an FSR = 17 (see Figure 9b). For the #10 MZI, we obtained a *V* = 0.34 and the FWHM = 3 nm with an FSR = 8 nm; see Figure 9c. Figure 9d shows the spatial frequency of MZIs with lengths of L = 2.5 cm, L = 4 cm, and L = 5 cm. The peak dominant intensity at zero is the core mode, and the dominant core cladding modes are localized in p≈21 nm, p≈15 nm, and p≈12 nm, respectively. It can also be seen that there were weak peaks that corresponded to higher-order cladding modes, but their contribution to the interference pattern emission was null. Therefore, the interference emission was generated by core and cladding modes with different spatial frequencies because the length L was not the same at each interferometer.

## 4. Discussion

Table 3 shows a comparison of the reported fiber interferometer structure in infrared and visible regions, as well as the range of core-offset displacement from 5 μm to 40 μm with SMF-MZI MZI [3,4]. The taper MZI structure [7,18] is shown in the infrared region and the microcavity MZI structure in the visible region. In [4], it is reported how a large lateral core-offset displacement affected the relative direction of the joints of two segments of SMF in the interference performance of an interferometer, and we obtained a visibility of *V =* 0.2 with displacements of 6 μm and 40 μm. The best interference pattern for the core-offset MZIs was obtained at 4.6 μm, and in other displacements, the visibility of the fringes decreased considerably, up to 50%, *V* = 0.12 for #4 MZI, *V* = 0.1 for #6 MZI, and *V* = 0.08 for #7 MZI. Alternatively, it reached almost zero; see Figure 8a,b. Therefore, it is important to measure the displacement in the joints of two segments of fiber to fabricate an interferometer based on SMF-28 in the visible region, and a minimal increase in the displacement can cause the interference emission to be lost. An optical fiber sensor with symmetric and asymmetric taper MZI structures and visibility up of *V* = 0.2 was reported. In [20], a visibility of *V* = 0.5 was reported in an optical fiber SM-800 and SM-600, and in [22], a V = 0.08 with MNF was described. The proposed taper MZI with an asymmetric relation waist length of 10–20 µm had a sinusoidal profile of interference emission with a *V* = 0.14 and an FSR = 23 nm. These parameters are within the visibility range of 0.1≤V≤1, [1,23]. In this study, an SMF-28 with an LED source was used, and the visibility values obtained were *V* = 0.14 for the taper MZI and *V* = 0.3 for the core-offset MZI.

### Sensing Application

A setup based on the MZI have been proposed for temperature measurement using fiber SMF [5,19]. It had a sensibility of 0.04 dBm/°C and was coated with aluminum with a sensibility of 120 pm/°C [5], and used doped fiber peanut structure with a sensibility of 0.158 nm/°C [19]. The core-offset MZI with a length of 2.5 cm was implemented for the experimental setup shown in Figure 3. The sensor was fixed on a hot plate (Thermo Scientific Cimarec, Waltham, MA, USA, Mod. SP131015) and the temperature response was measured using a Bosch GIS 500 temperature detector (with a temperature range of −30 °C to +500 °C and a resolution of ±1.8 °C). Figure 10a shows the temperature response that was detected in the range of 50 °C to 300 °C. A variation in the intensity amplitude interference was observed when the temperature increased; this was because, as the temperature increased, the effective refractive index of both the cladding and core modes increased [5]. The peaks and dips A, C, and E and B and D, respectively, were chosen to analyze the sensor’s sensitivity. The intensity amplitude variation versus temperature increase is shown in Figure 10b, and presented a good response to the change in temperature. Sensitivities of 1.13 a.u./°C, 1.58 a.u./°C, 1.77 a.u./°C, 3.53 a.u./°C, and 2.53 a.u./°C, with R^2^ values of 0.83077, 0.9538, 0.99393, 0.99769, and 0.97666, respectively, were achieved. The linear response to C, D was is better, and the best sensitivity and R^2^ were achieved for the dip D. To compare the sensibility obtained by our sensor with other reported manuscripts, the exit intensity was normalized, obtaining the best sensibility in the dip D of 1.0% every °C; see Figure 10c.

In Figure 10a, we can see that in the interference patterns (B, C, and D) and (A and E) were shifted to longer and shorter wavelength directions, respectively, when the temperature increased. This is because of the interference or resonant wavelength, expressed as [11,12,13,14]:(5)$$\lambda_{m}=\frac{2L_{0}\Delta n_{eff}}{2m+1}=\frac{2\left(\Delta n_{eff}^{m}+\beta \Delta T\right)\left(L_{0}+\alpha \Delta T\right)}{2m+1}$$
where m is an integer. The resonance wavelength shift versus temperature variation depended on the effective refractive indexes of the core and cladding modes. It depended on $2L_{0}\beta /\left(2m+1\right)$, where *β* is the thermo-optic coefficient difference core and cladding, and *α* is the thermal expansion coefficient. Thus, these corresponding changes can cause the interference dips to shift in the long or short wavelength direction in the transmission spectrum [14]. 

Figure 11 shows the sensibility of the sensor for the dip D; the sensitivity obtained was 20.3 pm/°C, with a R^2^ of 0.966604, in a temperature range of 50 °C to 150 °C. The other peaks and dips were analyzed, but the sensibility obtained was not good.

Table 4 shows a comparison of the temperature sensors reported in the infrared and the visible region. It can be observed that, in the sensor, a good sensibility of 20.3 pm/°C was obtained compared with [24,25,26,27] in the infrared region. In the comparative analysis of the visible region, a sensibility of 1% was obtained every °C, while in [28], sensibilities of 3.5%, 3%, and 1% is reported every °C in transmission. Reversible thermochromic micro-powders were used to add thermal sensing functionality into photocurable resin comprising polyhydroxyethyl methacrylate (p-HEMA)and polyethylene glycol diacrylate (PEGDA)-based polymer fibers.

The importance of the parameters of visibility range and FSR in interference emission is to determinate the sensibility of the interferometers for sensing applications. The sensor fabricated for the temperature obtained a good sensibility compared with those of the other configurations; therefore, it was expected to increase with a coating of a layer of metal on the sensor, e.g., gold [29], platinum [30], aluminum [5], or another a thin metal, to increase the sensitivity. This type of sensor with SMF and a white light source could have multiple measurement applications.

**Table 4 sensors-24-03026-t004:** Comparison between the sensibilities of the reported sensors.

Configuration	Range of Temperature	Sensitivity	Sensitivity Every 1 °C	Ref.
*Infrared region*				
Air cavities with capillary fiber between 2 SMFs	50 to 400 °C	0.8 pm/°C	-	[24]
SMF + hollow-core photonic crystal fiber (PCF)	17 to 900 °C	0.94 pm/°C	-	[25]
SMF + Hollow core tube + SMF	50 to 450 °C	0.902 pm/°C	-	[26]
SMF + NCF	100 to 700 °C	6.8 pm/°C	-	[27]
SMF + NCF (with a gold film) + SMF	20 to 80 °C	37.9 pm/°C	-	[31]
*Visible region*				
OF + polymer	25 to 35 °C	-	3.5%, 3% and 1%	[28]
Core-offset (SMF)	50 to 300 °C	-	1%	This work
Core-offset (SMF)	50 to 150 °C	20.3 pm/°C	-	This work

## 5. Conclusions

The visible light interference in the taper and core-offset MZIs structure was experimentally demonstrated using a standard SMF-28 for its fabrication. The visibility fringes of *V* = 0.14 were obtained with an FSR = 23 nm for the taper MZI structure. *V* = 0.3, *V* = 0.27, and *V* = 0.34 were obtained with FSR = 23 nm, 17 nm, and 8 nm using an MZI with lengths of L = 2 cm, L = 4 cm, and L = 5 cm, respectively. The core-offset MZI of 2.5 cm was characterized by a temperature sensor in a range from 50 °C to 300 °C. The best sensitivity obtained was 3.53 a.u./°C and 1% every °C in transmission, with R = 0.99769. The sensibility obtained was 20.3 pm/°C, with a R of 0.96604, in the temperature range from 50 °C to 150 °C. The MZIs based on this configuration could be used in the detection of other physical variables, such as the refractive index, strain, pressure, chemical properties, and biosensors.

## Figures and Tables

**Figure 1 sensors-24-03026-f001:**
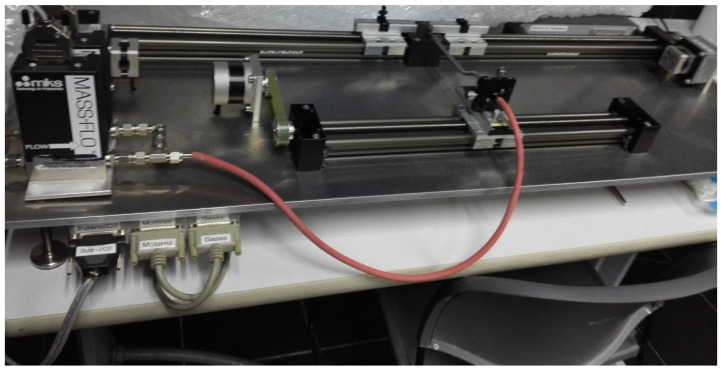
Stretching machine to fabricate the tapers (designed and built at laboratory of fiber optics, Universitat de València).

**Figure 2 sensors-24-03026-f002:**
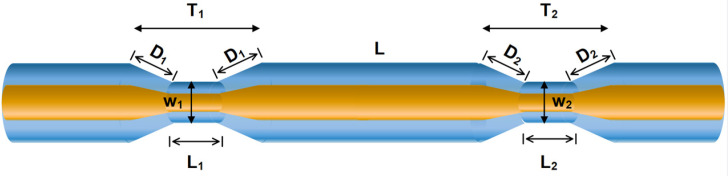
Schematic bi-taper MZI on SMF-28.

**Figure 3 sensors-24-03026-f003:**
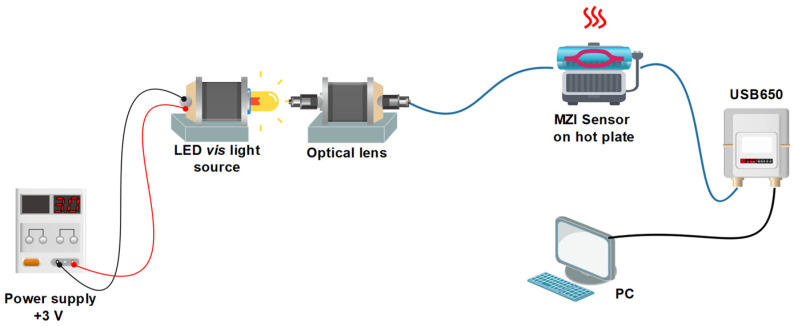
Experimental setup to measurement interference VIS pattern.

**Figure 4 sensors-24-03026-f004:**
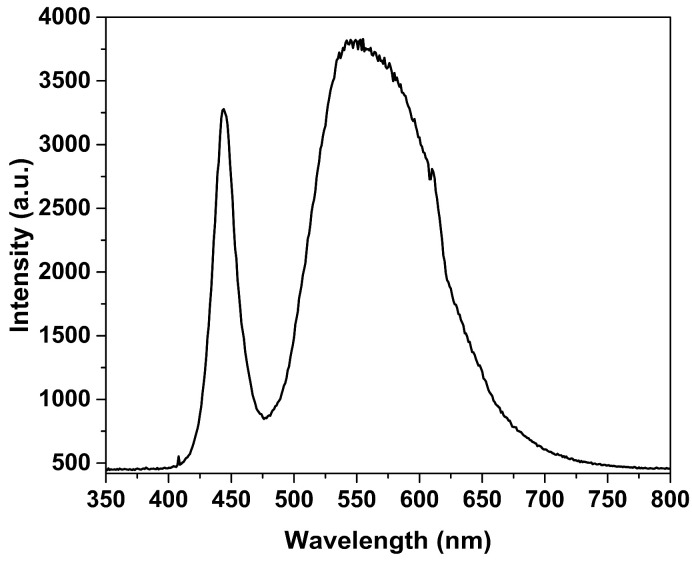
Spectral emission of LED source at room temperature.

**Figure 5 sensors-24-03026-f005:**
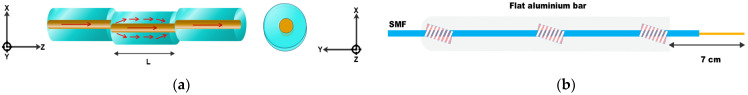
Structure of MZI core-offset splicing technique. (**a**) U type MZI structure; (**b**) fiber optics fixed on a flat aluminum bar.

**Figure 6 sensors-24-03026-f006:**
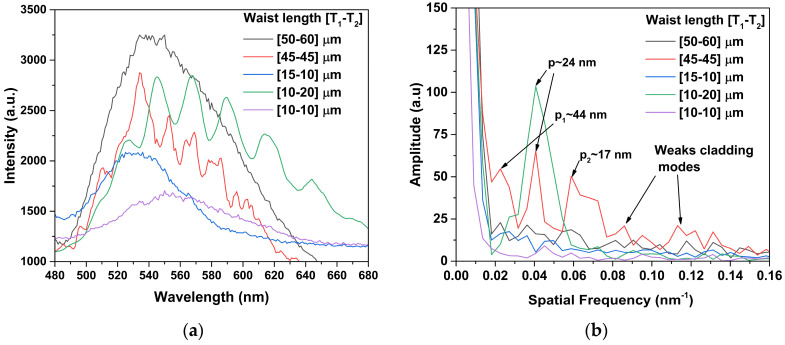
(**a**) Spectrum emission of #MZIs; (**b**) spatial frequency spectrum by FFT.

**Figure 7 sensors-24-03026-f007:**
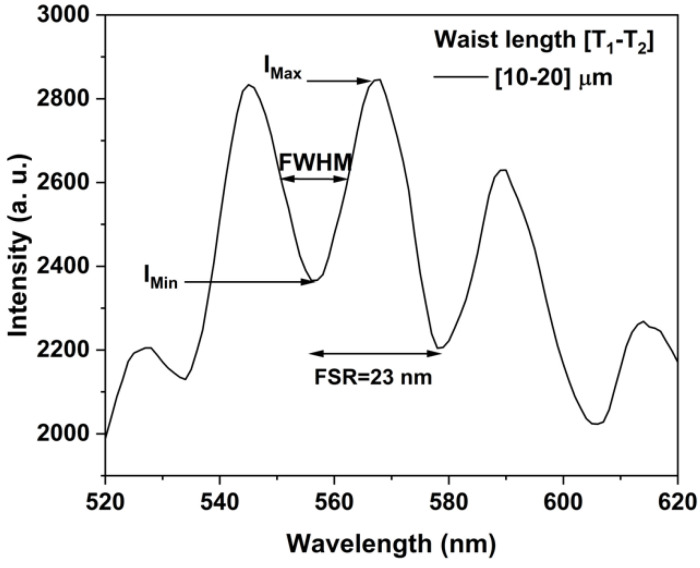
Transmission spectra of the taper MZI with waist length [10-20] μm.

**Figure 8 sensors-24-03026-f008:**
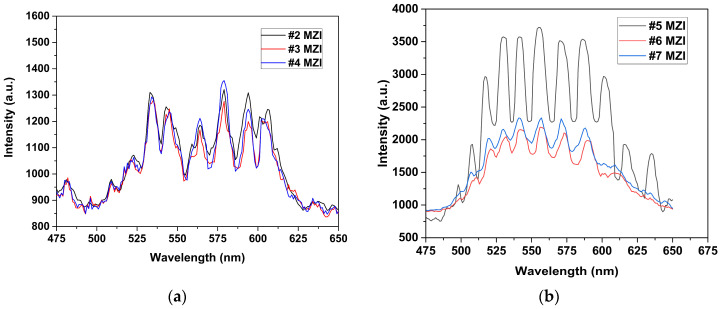
Spectrum emission of the core-offset MZI: (**a**) #2 MZI–#4 MZI, (**b**) #5 MZI–#7 MZI, (**c**) #8 MZI and #9 MZI. (**d**) Spatial frequency spectrum by FFT of the #2 MZI–#9 MZI, with a length of 4 cm.

**Figure 9 sensors-24-03026-f009:**
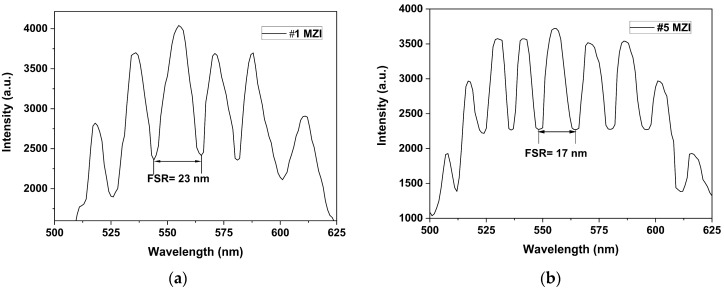
The pattern interference emissions: (**a**) #1 MZI with L = 2.5 cm; (**b**) #5 MZI with L = 4 cm; (**c**) #10 MZI with L = 5 cm; (**d**) the spatial frequency of the #1, #5, and # 10 MZIs.

**Figure 10 sensors-24-03026-f010:**
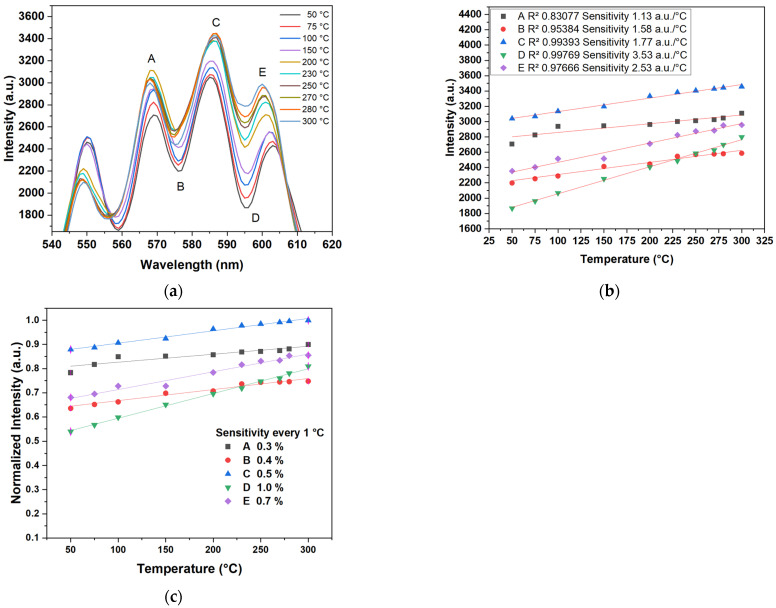
(**a**) Spectrum emission of temperature sensor. (**b**) Experimental and fitting results of the sensitivity. (**c**) Normalized intensity experimental and fitting results of sensitivity.

**Figure 11 sensors-24-03026-f011:**
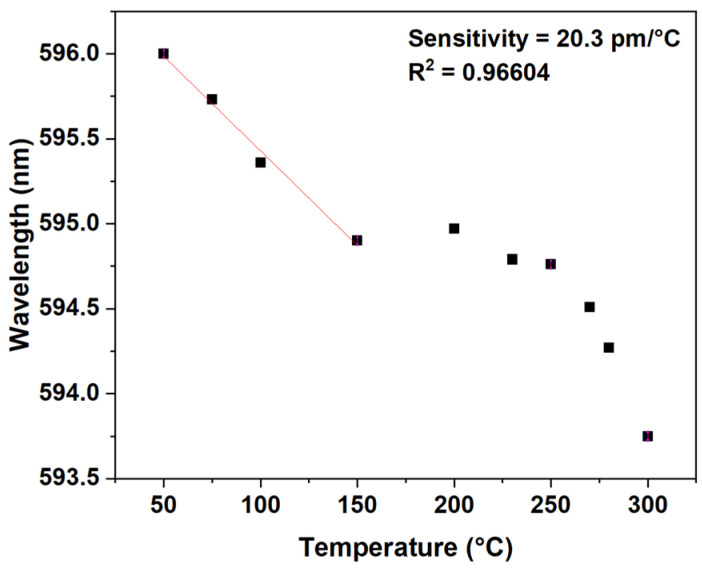
Sensor sensibility for the dip D.

**Table 1 sensors-24-03026-t001:** Parameters of symmetric and asymmetric tapered MZI.

#MZI	Taper 1 (T_1_)			Taper 2 (T_2_)		
Waist Diameter Relation [μm]	Waist Length [mm]	Waist Diameter [μm]	Length [mm]	Waist Length [mm]	Waist Diameter [μm]	Length [mm]
#[T1-T2]	L_1_	W_1_	D_1_	L_2_	W_2_	D_2_
1 [10-20]	5.05	10	2.52	7.33	20	3.66
2 [45-45]	6.13	45	3.06	6.13	45	3.06
3 [10-10]	5.05	10	2.52	5.05	10	2.52
4 [50-60]	1.83	50	0.91	29.35	60	14.67
5 [15-10]	8.48	15	4.24	5.05	10	2.52

**Table 2 sensors-24-03026-t002:** Parameters of symmetric and asymmetric SMF-SMF-SMF structure core-offset MZI.

#MZI	Length L [cm]	First Splice Displacement $\left[x_{1},y_{1}]\;\mathrm{Directions}\;[\mu m\right]$	Second Splice Displacement $\left[x_{2},y_{2}]\;\mathrm{Directions}\;[\mu m\right]$
1	2.5	[−4.6, 0.0]	[+4.6, 0.0]
2	4	[−3.0, 0.0]	[+3.0, 0.0]
3	4	[−3.0, 0.0]	[+4.0, 0.0]
4	4	[−4.0, 0.0]	[+4.0, 0.0]
5	4	[−4.6, 0.0]	[+4.6, 0.0]
6	4	[−5.0, 0.0]	[+4.0, 0.0]
7	4	[−5.0, 0.0]	[+4.5, 0.0]
8	4	[−6.0, 0.0]	[+5.0, 0.0]
9	4	[−6.0, 0.0]	[+6.0, 0.0]
10	5	[−4.6, 0.0]	[+4.6, 0.0]

**Table 3 sensors-24-03026-t003:** Comparison of the reported fiber interferometer structures in infrared and visible regions.

Interferometer Structure and Operation Region	Core-Offset [μm]	Interferometer Length (cm)	FSR (nm)	Visibility	Ref.
*Infrared region*					
Core-offset MZI (SMF)	5	4	12	0.1	[2]
Core-Offset MZI (SMF)	6 to 40	3	15	0.2	[3]
Core-offset (SMF-Al coated)	30	2	16	0.7	[4]
Taper MZI (SMF)	-	2	19	0.2	[6]
Taper MZI (EDF)	-	4.5	12	0.25	[13]
*Visible region*					
Microcavity MZI (SMF-800)	-	-	50	0.5	[15]
Microcavity (MNF-SMF)	-	-	8	0.09	[17]
Core-offset (SMF)	4.6	2.5	23	0.3	This work

## Data Availability

Data are contained within the article.

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
