# Peer review of "Experimental Study of White Light Interferometry in Mach–Zehnder Interferometers Based on Standard Single Mode Fiber"

_sensors, 2024, doi:10.3390/s24103026_

Round 1

Reviewer 1 Report

Comments and Suggestions for Authors

The authors aim to introduce a cost-effective sensor based on Mach-Zehnder Interferometer (MZI). To achieve this, they experimentally demonstrated the interference pattern within the visible (VIS) region. Two MZIs were fabricated using a structure of Single Mode Fiber (SMF)-SMF-SMF. The first interferometer was constructed using a bi-taper technique, while the second was based on a core-offset splicing technique. Both MZIs were fabricated using standard optical fiber (SMF-28), and their interference emissions were analyzed individually. Prior to acceptance for publication in Sensors, I kindly request the authors to consider several comments and suggestions.

1.     The abstract appears to serve more as an introduction rather than encapsulating the essence of the manuscript. It is recommended to revise it for improved readability. The abstract should be adjusted to enhance comprehension of the manuscript/project's objectives. From what I gather, the authors emphasize the primary objective as creating a "low-cost" sensing system applicable to industrial and medical settings.

2.     The narrative in English requires significant improvement. Upon reading the manuscript, it's evident that the flow is disjointed and could benefit from refinement. Strengthening the coherence and cohesion of the language would enhance the readability and overall quality of the manuscript.

3.     The quality of the literature review needs significant improvement. Many statements presented in the introduction lack proper references to support them. To mention few, for instance, the following sentences lack appropriate backing from references.

Page 1 “The optical fiber sensor based on in line Mach-Zehnder interferometer (MZI) have become more attractive research.”

4.     Figure 8 references additional comparison studies, yet it's challenging to grasp the significance of references such as "#2 MZI" or "#3 MZI." It's advisable to provide clearer explanations when referring to specific samples or data in order to enhance the clarity and understanding of the outcomes.

5.     On page 10, the authors mention "some setup," but it's unclear what exactly they are alluding to. It would be beneficial to elaborate on this reference for better comprehension or modify the sentence as "some setup" doesn't flow well or doesn't seem to be scientific in nature.

Comments on the Quality of English Language

The narrative in English requires significant improvement. Upon reading the manuscript, it's evident that the flow is disjointed and could benefit from refinement. Strengthening the coherence and cohesion of the language would enhance the readability and overall quality of the manuscript.

Author Response

REVISOR 1

Your review of the manuscript is appreciated, the observations have been taken into consideration and revised below:

  1. The abstract appears to serve more as an introduction rather than encapsulating the essence of the manuscript. It is recommended to revise it for improved readability. The abstract should be adjusted to enhance comprehension of the manuscript/project's objectives. From what I gather, the authors emphasize the primary objective as creating a "low-cost" sensing system applicable to industrial and medical settings.

R: The abstract has been modified following your recommendations

In this work, we experimentally analyzed and demonstrated the performance of an in-line Mach-Zehnder interferometers in the visible region, with LED light source. The different waist diameter taper and asymmetric core-offset interferometers proposed used a single-mode fiber (SMF). The visibility achieved is V=0.14 with an FSR of 23 nm, for the taper MZI structure and a visibility of V= 0.3, V=0.27 and V=0.34 with an FSR of 23 nm, 17 nm and 8 nm and with a separation length L of 2.5 cm, 4.0 cm and 5.0 cm between the core-offset structure, respectively. The experimental investigation of the response to the temperature sensor ranging from 50 °C to 300 °C, the sensitivity obtained was 3.53 a.u./°C, with  of 0.99769 and 1 % every 1 °C in the transmission. For a range of 50 °C to 150 °C, 20.3 pm/°C with a  of 0.96604, was obtained.

  1. The narrative in English requires significant improvement. Upon reading the manuscript, it's evident that the flow is disjointed and could benefit from refinement. Strengthening the coherence and cohesion of the language would enhance the readability and overall quality of the manuscript.

R: Your observations are appreciated and enhance the manuscript. The redaction was revised and modified for a better coherence and cohesion in the language.

  1. The quality of the literature review needs significant improvement. Many statements presented in the introduction lack proper references to support them. To mention few, for instance, the following sentences lack appropriate backing from references.

Page 1 “The optical fiber sensor based on in line Mach-Zehnder interferometer (MZI) have become more attractive research.”

R: Your comment is appreciated. The references have been added to the introduction.

“The optical fiber sensor based on in line Mach-Zehnder interferometer (MZI) have become more attractive research…” [2]

[2] Xia Hao, Zhengrong Tong, Weihua Zhang, Ye Cao,A fiber laser temperature sensor based on SMF core-offset structure, Optics Communications,Volume 335, 2015,Pages 78-81,ISSN 0030-4018, ttps://doi.org/10.1016/j.optcom.2014.08.065.

  1. Figure 8 references additional comparison studies, yet it's challenging to grasp the significance of references such as "#2 MZI" or "#3 MZI." It's advisable to provide clearer explanations when referring to specific samples or data in order to enhance the clarity and understanding of the outcomes.

R: The explanation has been modified for a better comprehension.

The core-offset technique was studied experimentally with different displacement in the  axis, in both the first and the second fusion splice to fabricate the MZI keeping the length L at 4 cm.  First, the MZI numbered from #2 to #9 of the Table 2, were analyzed. The Figure 8(a) to 8(c) shows the evolution of interference spectrum dependent on the displacement of the  axis in splicing fiber. We can see that the extinction ratio (ER) increases or decreases as the displacement [x,y] increases; of the MZIs.

The spatial frequency of the MZIs mentioned is plotted in Figure 8(d), for the MZI (#5) a strong cladding mode is present in  and is important in contributing significantly to emission pattern interference; therefore, the interference pattern is due to the fundamental core and cladding mode in . For the MZIs (#6) and (#7), the amplitude of the spatial frequency in   is smaller by 75% proximally than for the MZI (#5), but the output pattern also has a sinusoidal form, as can see in the Figure 8(b).

We can see that, the interference patterns for another core-offset MZIs, are not completely sinusoidal waveform, due to that it is formed by fundamental core mode and a number of high-order cladding modes. This is generated, when the white light is launched into the an SMF, higher-order modes (LP0m) are excited [11], [12]. It is observed in all MZIs that the interference pattern is generated by the fundamental core mode and of high-order cladding modes due to SMF fiber has a performance multimode, except the MZI (#5), in which it is observed a high-order cladding modes and higher order modes are not predominant due to their weakness. Finding, that the optimal displacement in the  axis for the MZI is [-4.6,0.0] μm and [+4.6,0.0] μm, for the first and second junction splice, respectively.

  1. On page 10, the authors mention "some setup," but it's unclear what exactly they are alluding to. It would be beneficial to elaborate on this reference for better comprehension or modify the sentence as "some setup" doesn't flow well or doesn't seem to be scientific in nature.

R: The explanation has been enhanced with the references.  

Some setup based on the MZI have been proposed for temperature measurement, using fiber SMF [5], [19] with a sensibility of 0.04 dBm/ °C and coated with aluminum with a sensibility of 120 pm/°C [5], using doped fiber peanut structure with a sensibility of 0.158 nm/°C [19], were reported .

Reviewer 2 Report

Comments and Suggestions for Authors

As the authors have already shown along the text and in detail in Table 3, their interferometer is not new. Therefore: What is the most important new result obtained by the authors? What have the authors done that has not been done and reported before?

At the beginning of the article the authors mention several promising applications however along the article they discuss none in detail, except  the one dealing with this interferometer as a temperature sensor, which is something not new at all.

In it's present form my recommendation is to REJECT this article.

A last, rather intriguing, comment is; What exactly was the contribution to this article by a coauthor from the Business and Accounting Faculty of a University. I suggest to display the precise contribution of each author to this paper.

Comments on the Quality of English Language

No comments

Author Response

REVISOR 2

As the authors have already shown along the text and in detail in Table 3, their interferometer is not new. Therefore: What is the most important new result obtained by the authors? What have the authors done that has not been done and reported before?

At the beginning of the article the authors mention several promising applications however along the article they discuss none in detail, except the one dealing with this interferometer as a temperature sensor, which is something not new at all.

A last, rather intriguing, comment is; What exactly was the contribution to this article by a coauthor from the Business and Accounting Faculty of a University. I suggest to display the precise contribution of each author to this paper.

R: Thank you for your revision and the observations. This manuscript is the experimental analysis of the interferometers Mach-Zehnder, using a SMF and a white light source. The interference in white light with SMF are known to be obtained with noise interference, in this work the sinusoidal interference was obtained with a better visibility.

The section 4.1 Sensing Application.

In which the response of the MZI with respect to the increase in temperature is reported. According to the reviewers' recommendation, the analysis of spectrum transmission and the shift in wavelength in response to temperature is added. In this section the main contributions and results are expanded.

4.1. Sensing Application

Some setup based on the MZI have been proposed for temperature measurement, using fiber SMF [5], [19] with a sensibility of 0.04 dBm/ °C and coated with aluminum with a sensibility of 120 pm/°C [5], using doped fiber peanut structurewith a sensibility of 0.158 nm/°C [19], were reported. The core-offset MZI with length of 2.5 cm, was implemented for the experimental setup of the Figure 3. The sensor was fixed on a hot plate (Thermo Scientific Cimarec, Mod. SP131015) and the temperature response was measured by temperature detector Bosch GIS 500 (with a temperature range of  to  and a resolution of ). The Figure 10(a), shows the temperature response that was detected in the range of  to . It was observed a variation of the intensity amplitude interference when the temperature increased, this is because, as the temperature increases, the effective refractive index of both the cladding and core modes increases [5]. The peaks and dips A, C, E and B, D, respectively; were chosen to analyze the sensor sensitivity. The intensity amplitude variation versus temperature increased is shown in the Figure 10(b), present a good response to the change temperature. Sensitivities of 1.13 a.u./°C, 1.58 a.u./°C, 1.77 a.u./°C, 3.53 a.u./°C and 2.53 a.u./°C; with R² of 0.83077, 0.9538, 0.99393, 0.99769 and 0.97666, respectively, are achieved. The linear response is better to C, D and the best sensitivity and R², is the dip D. To compare the sensibility obtained by our sensor with other reported manuscripts, the exit intensity is normalized, obtaining the best sensibility in the dip D of 1.0% every °C, Figure 10c.

(a)

(b)

(c)

Figure 10. (a) Spectrum emission of temperature sensor. (b) Experimental and fitting results of the sensitivity. (c) Normalized intensity experimental and fitting results of sensitivity.

In the Figure 10a, we can see that in the interference pattern (B, C and D) and (A and E) were shifted to longer and shorter wavelength direction, respectively; when the temperature increased. This is because the interference  or resonant wavelength, expressed as [10]-[12]:

(5)

where  is an integer.  The resonance wavelength shift versus temperature variation, depends of the effective refractive indexes of core and cladding modes. It depends by  , where b  is the thermo-optic coefficient difference core and cladding, and a is the thermal expansion coefficient. Thus, these corresponding changes can originate the interference dips shift to long or short wavelength direction in the transmission spectrum [14].

The Figure 11, shows the sensibility of the sensor for the dip D, the sensitivity obtained was at 20.3 pm/°C with a R2 of 0.966604, in a temperature range of 50 °C to 150 °C. The others peaks and dips where analyzed, but the sensibility obtained was not good.

Figure 11. Sensor sensibility for the dip D.

The table 4, shows a comparison of the temperature sensors reported in the infrared and the visible region. It can be observed that in the sensor it was obtained a good sensibility of 20.3 pm/°C, compared with [24-27], in the infrared region. In the comparative analysis of the visible region, it was obtained a sensibility of 1% every °C, while in [28] is reported a sensibility of 3.5%, 3% and 1% every °C, in transmission, using a reversible thermochromic micro-powders were used to add thermal sensing functionality into photocurable resin comprised of polyhydroxyethyl methacrylate(p-HEMA)and polyethylene glycol diacrylate (PEGDA) based polymer fibers.

The importance of the parameter of visibility range and the FSR in the interference emission is to determinate its sensibility in the interferometers for the sensing application. The sensor fabricated for the temperature obtained a good sensibility compared with the other configurations, therefore it was an expected to increase with a coating a layer of metal on the sensor, such as gold [29], platinum [30] , aluminum [5] or another a thin metal  to increase their sensitivity. This type of sensor with SMF and a white light source could have multiple measurement applications.

Table 4. Comparison between the sensibility of the reported sensor.

Configuration

Range of temperature

Sensitivity

Sensitivity every 1 °C

Ref.

Infrared region

Air cavities with capillary fiber between 2 SMFs

50 to 400 °C

0.8 pm/°C

-

[24]

SMF + hollow-core photonic crystal fiber (PCF)

17 to 900 °C

0.94 pm/°C

-

[25]

SMF + Hollow core tube + SMF

50 to 450 °C

0.902 pm/°C

-

[26]

SMF + NCF

100 to 700 °C

6.8 pm/°C

-

[27]

SMF + NCF (with a gold film) + SMF

20 to 80 °C

37.9 pm/°C

-

[31]

Visible region

OF + polymer

25 to 35 °C

-

3.5%, 3% and 1%

[28]

Core-offset (SMF)

50 to 300 °C

-

1%

This work

Core-offset (SMF)

50 to 150 °C

20.3 pm/°C

-

This work

  1. Conclusions

It has been experimentally demonstrated the visible light interference in the taper and core-offset MZIs structure, using a standard SMF-28 for its fabrication. The obtained visibility fringes of V=0.14 and with an FSR= 23 nm for the taper MZI structure and with V=0.3, V=0.27 and V=0.34 with an FSR= 23 nm, 17 nm and 8nm, using a MZI with a length of L=2 cm, L= 4 cm and L= 5 cm, respectively. The core-offset MZI of 2.5 cm, was characterized by temperature sensor in a range from 50 °C to 300 °C. The best sensitivity obtained was 3.53 a.u./°C and 1% every °C in transmission, with a R= 0.99769. The sensibility obtained was of 20.3 pm/°C with a R of 0.96604 in a temperature range from 50 °C to 150 °C. The MZIs based in this configuration, could be used in the detection of other physical variables, as in refractive index, strain, pressure, chemical and biosensors.

In attention to your comment is; “What exactly was the contribution to this article by a coauthor from the Business and Accounting Faculty of a University. I suggest to display the precise contribution of each author to this paper.

The co-autor, belonging to the Business and Accounting Faculty is a computer science graduate and his line of work has been the optic fibers used in telecommunications. He is currently subscribed to the Business and Accounting Faculty and not the Exact Science department but that is due to internal procedures in the university, however as mentioned before he collaborates with the academic team for research purposes.

Methodology, J. L. Cano-Perez, J. Gutierrez-Gutiérrez, C. Perezcampos-Mayoral, E. Perez-Campos, M. S. Pina-Canseco, L. Tepech-Carrillo and E. I. Guerra-Hernandez;

Validation, M. S. Pina-Canseco and L. Tepech-Carrillo;

Formal analysis, M. Vargas-Treviño;

Investigation, J. L. Cano-Perez, J. Gutierrez-Gutiérrez, C. Perezcampos-Mayoral and E. I. Guerra-Hernandez; Writing – original draft, J. L. Cano-Perez and J. Gutierrez-Gutiérrez;

Writing – review & editing, J. L. Cano-Perez, J. Gutierrez-Gutiérrez, J. M. Sierra-Hernandez and R. Rojas-Laguna; Visualization, C. Perezcampos-Mayoral, A. Martinez-Helmes;

Supervision, J. Gutierrez-Gutiérrez, E. Perez-Campos, J. M. Estudillo-Ayala, J. M. Sierra-Hernandez and R. Rojas-Laguna;

Project administration, A. Martinez-Helmes.

Reviewer 3 Report

Comments and Suggestions for Authors

The manuscript under consideration is devoted to the experimental investigation of the emission spectra of white LED transmitted through the interferometers made of single mode fiber. The manuscript is worth written and also has a questionable content. My comments are presented below.

1) In the «Introduction» section authors use «far-infrared (FIR)» term (see line 57) citing Refs. [1-9, 14-18]. The use of term far-infrared is inappropriate, because typically FIR spectral range corresponds to the wavelength longer than 20 um, whereas in abovementioned Refs. cited in the manuscript only near-infrared spectral range is investigated (wavelength ~ 1500 nm).

2) In the “2.1 Design and fabrication of taper MZI” authors declare that they use single mode fiber (SMF-28 of Thorlabs), which is optimized for the operation in the wavelength range 1260-1650 nm (see Line 81). However, during the experimental investigations, authors use white LED as a light source and provide measurements in the wavelength range 400-700 nm. In this case, one can see that the fiber SMF-28 is not suitable for the visible spectral range. Moreover, SMF-28 optical fiber apparently is not single mode for the visible range photons due to the increased mode field diameter and fiber core diameter as well. Thus, the research carried out by the authors raises doubts.

3) In the Line 96 designation Ddown2 has an error – it should be Dup2.

4) Figure 4 shows emission spectrum of the white LED. It is not clear in what conditions it was measured, what was the experimental setup for this measurement. Moreover, the emission spectrum of the LED shows the emission intensity ~500 a.u. for the wavelength shorter 400 nm and longer 750 nm. The nature of this emission is unclear.

5) In the Line 126 authors declare that Spectrum Analyzer had a “slit of 5 in the software configuration of the computer”. This information is meaningless, because it can’t help to the reader obtain spectral resolution of the experimental setup. The real width of the slit of the spectrum analyzer or spectral resolution of the experimental setup must be mentioned in the manuscript.

6) In table 1 as well as in the text of the manuscript (for example, Line 133) authors present waist length with a very high accuracy, up to 1 nm. It is difficult to believe that authors can make a such precision fiber production.

7) In the emission spectra, presented in Figs. 6-9, authors show spectra in narrow spectral range (~475-680 nm or even narrower), whereas white LED (as it is shown in Fig. 4) has a broader emission spectrum (400-750nm). Thus, emission spectra, measured in a wide spectral range, corresponded to the LED emission spectrum, should be presented.

8) The term FFT is often misspelled FTT (see, Lines 193,303 for example).

9) In Line 304 authors, describing Fig. 10a, declare that “interference pattern was shifted a longer wavelength direction when the temperature increased”. However, in Line 310 authors declare that “the interference patterns redshifts as the temperature increase”. The abovementioned expressions contradict each other. Moreover, one can see that in Fig. 10a different extremes in the emission spectrum have a different shift (red or blue) with the temperature increase (for example, peak E redshifts and peak C blueshifts).

10) The legend in Fig. 10b has designation of emission peaks (A,a,B,b,C) that differs from the designation presented in Fig. 10a (A,B,C,D,E).

11) Finally, authors present sensing application in Section 4.1 corresponded to temperature sensing using MZIs. This idea looks questionable, because a weak dependence of the peaks intensities on the temperature is observed and it also requires spectral measurements for temperature sensing.

Comments on the Quality of English Language

The manuscript contains a large number of typos, incorrectly constructed sentences, etc. A good language correction is needed.

Author Response

REVISOR 3

  • In the «Introduction» section authors use «far-infrared (FIR)» term (see line 57) citing Refs. [1-9, 14-18]. The use of term far-infrared is inappropriate, because typically FIR spectral range corresponds to the wavelength longer than 20 um, whereas in abovementioned Refs. cited in the manuscript only near-infrared spectral range is investigated (wavelength ~ 1500 nm).

R: The comment is correct, thank you. The amendment has been made in the manuscript.

57        gas sensing [16]. The optical fiber MZIs have been studied main in the far-infrared (FIR)

57        gas sensing [16]. The optical fiber MZIs have been studied mainly in the infrared (IR)

2) In the “2.1 Design and fabrication of taper MZI” authors declare that they use single mode fiber (SMF-28 of Thorlabs), which is optimized for the operation in the wavelength range 1260-1650 nm (see Line 81). However, during the experimental investigations, authors use white LED as a light source and provide measurements in the wavelength range 400-700 nm. In this case, one can see that the fiber SMF-28 is not suitable for the visible spectral range. Moreover, SMF-28 optical fiber apparently is not single mode for the visible range photons due to the increased mode field diameter and fiber core diameter as well. Thus, the research carried out by the authors raises doubts.

R: Your comment is correct, “the optic fiber SMF-20 is not a single mode for the photons in the visible range due to the bigger diameter from mode field in addition to the diameter from the fiber nucleus.

In response to your recommendation, in page 8 the explanation has been expanded referencing figure 8.

… We can see that, the interference patterns for another core-offset MZIs, are not completely sinusoidal waveform, due to that it is formed by fundamental core mode and a number of high-order cladding modes. This is generated, when the white light is launched into the an SMF, higher-order modes (LP0m) are excited [10], [11].It is observed in all MZIs that the interference pattern is generated by the  fundamental core and the high-order cladding modes due to SMF fiber has a performance multimode, except the MZI (#5), in which it is observed a strong cladding mode and high-order cladding modes are not predominant due to their weakness. Finding, that the optimal displacement in the  axis for the MZI is [-4.6,0.0] μm and [+4.6,0.0] μm, for the first and second junction splice, respectively. …

3) In the Line 96 designation Ddown2 has an error – it should be Dup2.

R: The comment is correct, thank you. The amendment has been made in the manuscript.

96            length (L1 and L2), waist diameter (W1 and W2) and up-taper (Dup1 and Ddown2), respectively;

96            length (L1 and L2), waist diameter (W1 and W2) and up-taper (Dup1 and Dup2), respectively;

4) Figure 4 shows emission spectrum of the white LED. It is not clear in what conditions it was measured, what was the experimental setup for this measurement. Moreover, the emission spectrum of the LED shows the emission intensity ~500 a.u. for the wavelength shorter 400 nm and longer 750 nm. The nature of this emission is unclear.

R: The white led is measured at room temperature, without configuration to the fiber SMF-28, in figure 4 the spectrum is from 400nm to 750nm, from figure 6 the spectrum range is reduced approximately from 480 to 650nm due to better analysis and pattern visualization from interference from the MZIs.

Added to page 4:

… with a wavelength range of 400 nm to 650 nm (the spectral emission at room temperature is shown in Figure 4).

In figure 4:

Figure 4. Spectral emission of LED Source, at room temperature.

5) In the Line 126 authors declare that spectrum Analyzer had a “slit of 5 in the software configuration of the computer”. This information is meaningless, because it can’t help to the reader obtain spectral resolution of the experimental setup. The real width of the slit of the spectrum analyzer or spectral resolution of the experimental setup must be mentioned in the manuscript.

R: Thank you for your comment, the width of the slit has been added.

147          is obtained by Spectrum Analyzer (Ocean Optics, USB650) with a slit of 5 in the software

147          is obtained by Spectrum Analyzer (Ocean Optics, USB650) with a slit of 5  in the software

6) In table 1 as well as in the text of the manuscript (for example, Line 133) authors present waist length with a very high accuracy, up to 1 nm. It is difficult to believe that authors can make a such precision fiber production.

R: Thank you, your comment enhance the manuscript, there was an error in the units used in the table chart as well as in the line 133. The units have been modified to two decimals.

The table 1 and the manuscript have been updated.

#MZI

Taper 1 (T1)

Taper 2 (T2)

Waist diameter relation

[]

Waist length

[mm]

Waist diameter [

Length

[mm]

Waist length

[mm]

Waist diameter []

Length

[mm]

#[T1-T2]

L1

W1

D1

L2

W2

D2

1 [10-20]

5.05

10

2.52

7.33

20

3.66

2 [45-45]

6.13

45

3.06

6.13

45

3.06

3 [10-10]

5.05

10

2.52

5.05

10

2.52

4 [50-60]

1.83

50

0.91

29.35

60

14.67

5 [15-10]

8.48

15

4.24

5.05

10

2.52

133          taper is kept to   and waist length between 1.833 µm to 29.359 µm, …

133          taper is kept to   and waist length between 1.83 mm to 29.35 mm, …

7) In the emission spectra, presented in Figs. 6-9, authors show spectra in narrow spectral range (~475-680 nm or even narrower), whereas white LED (as it is shown in Fig. 4) has a broader emission spectrum (400-750nm). Thus, emission spectra, measured in a wide spectral range, corresponded to the LED emission spectrum, should be presented.

R: Your observations are appreciated. The emission spectra from figure 4 has a range from 400-7500nm approximately. In figures 6-9, the spectrum range is narrower, 475-680nm approximately because the interference patter is more significant in this range for our analysis. This was the reason the complete spectral range was not presented.

8) The term FFT is often misspelled FTT (see, Lines 193,303 for example).

R: Your observations are appreciated, the misspelled has been modified in figure 6. The description from figure 10 will be changed completely.

193      Figure 6. (a) Spectrum emission of #MZIs; (b) Spatial frequency spectrum by FTT.

193      Figure 6. (a) Spectrum emission of #MZIs; (b) Spatial frequency spectrum by FFT.

9) In Line 304 authors, describing Fig. 10a, declare that “interference pattern was shifted a longer wavelength direction when the temperature increased”. However, in Line 310 authors declare that “the interference patterns redshifts as the temperature increase”. The abovementioned expressions contradict each other. Moreover, one can see that in Fig. 10a different extremes in the emission spectrum have a different shift (red or blue) with the temperature increase (for example, peak E redshifts and peak C blueshifts).

R: The comment is correct, thank you. The amendment has been made in the manuscript.

In the Figure 10, we can see that in the interference pattern (B, C and D) and (A and E) were shifted to longer and shorter wavelength direction, respectively; when the temperature increased. This is because the interference  or resonant wavelength, expressed as [10]-[12]:

where  is an integer.  The resonance wavelength shift versus temperature variation, depends of the effective refractive indexes of core and cladding modes. It depends by  , where b  is the thermo-optic coefficient difference core and cladding, and a is the thermal expansion coefficient. Thus, these corresponding changes can originate the interference dips shift to long or short wavelength direction in the transmission spectrum. [13]

10) The legend in Fig. 10b has designation of emission peaks (A,a,B,b,C) that differs from the designation presented in Fig. 10a (A,B,C,D,E).

R: Thank you for your comment, the image 10b has been modified

11) Finally, authors present sensing application in Section 4.1 corresponded to temperature sensing using MZIs. This idea looks questionable, because a weak dependence of the peaks intensities on the temperature is observed and it also requires spectral measurements for temperature sensing.

R: Thank you for your comments. The spectral analysis from the measurement of the temperature and a comparative table is added with the reported sensors.

To compare the sensibility obtained by our sensor with other reported manuscripts, the exit intensity is normalized, obtaining the best sensibility in the dip D of 1.0% every °C, Figure 10c.

Figure 10c. Normalized intensity experimental and fitting results of sensitivity.

The Figure 11, shows the sensibility of the sensor for the dip D, the sensitivity obtained was at 20.3 pm/°C with a R2 of 0.966604, in a temperature range of 50 °C to 150 °C. The others peaks and dips where analyzed, but the sensibility obtained was not good.

Figure 11. Sensor sensibility for the dip D.

The table 4, shows a comparison of the temperature sensors reported in the infrared and the visible region. It can be observed that in the sensor it was obtained a good sensibility of 20.3 pm/°C, compared [24-27], in the infrared region. In the comparative analysis of the visible region, it was obtained a sensibility of 1% every °C, while in [29] is reported a sensibility of 3.5%, 3% and 1% every °C, in transmission, using a reversible thermochromic micro-powders were used to add thermal sensing functionality into photocurable resin comprised of polyhydroxyethyl methacrylate(p-HEMA)and polyethylene glycol diacrylate (PEGDA) based polymer fibers.

The importance of the parameter of visibility range and the FSR in the interference emission is to determinate its sensibility in the interferometers for the sensing application. The sensor fabricated for the temperature obtained a good sensibility compared with the other configurations, therefore it was an expected to increase with a coating a layer of metal on the sensor, such as gold [23], platinum [24] , aluminum [4] or another a thin metal  to increase their sensitivity. This type of sensor with SMF and a white light source could have multiple measurement applications.

Table 4. Comparison between the sensibility of the reported sensor

Configuration

Range of temperature

Sensitivity

Sensitivity every 1 °C

Ref.

Infrared region

Air cavities with capillary fiber between 2 SMFs

50 to 400 °C

0.8 pm/°C

-

[24]

SMF + hollow-core photonic crystal fiber (PCF)

17 to 900 °C

0.94 pm/°C

-

[25]

SMF + Hollow core tube + SMF

50 to 450 °C

0.902 pm/°C

-

[26]

SMF + NCF

100 to 700 °C

6.8 pm/°C

-

[27]

SMF + NCF (with a gold film) + SMF

20 to 80 °C

37.9 pm/°C

-

[28]

Visible region

OF + polymer

25 to 35 °C

-

3.5%, 3% and 1%

[29]

Core-offset (SMF)

50 to 300 °C

-

1%

This work

Core-offset (SMF)

50 to 150 °C

20.3 pm/°C

-

This work

  1. Conclusions

It has been experimentally demonstrated the visible light interference in the taper and core-offset MZIs structure, using a standard SMF-28 for its fabrication. The obtained visibility fringes of V=0.14 and with an FSR= 23 nm for the taper MZI structure and with V=0.3, V=0.27 and V=0.34 with an FSR= 23 nm, 17 nm and 8nm, using a MZI with a length of L=2 cm, L= 4 cm and L= 5 cm, respectively. The core-offset MZI of 2.5 cm, was characterized by temperature sensor in a range from 50 °C to 300 °C. The best sensitivity obtained was 3.53 a.u./°C and 1% every °C in transmission, with a R= 0.99769. The sensibility obtained was of 20.3 pm/°C with a R of 0.96604 in a temperature range from 50 °C to 150 °C. The MZIs based in this configuration, could be used in the detection of other physical variables, as in refractive index, strain, pressure, chemical and biosensors.

Round 2

Reviewer 2 Report

Comments and Suggestions for Authors

None